# Further Studies on the 3-Ketosteroid 9α-Hydroxylase of *Rhodococcus ruber* Chol-4, a Rieske Oxygenase of the Steroid Degradation Pathway

**DOI:** 10.3390/microorganisms9061171

**Published:** 2021-05-29

**Authors:** Sara Baldanta, Juana María Navarro Llorens, Govinda Guevara

**Affiliations:** Department of Biochemistry and Molecular Biology, Universidad Complutense de Madrid, c/Jose Antonio Novais 12, 28040 Madrid, Spain; sabaldan@ucm.es

**Keywords:** 3-ketosteroid 9α-hydroxylase, steroids, *Rhodococcus ruber*, AD, ADD

## Abstract

The biochemistry and genetics of the bacterial steroid catabolism have been extensively studied during the last years and their findings have been essential to the development of biotechnological applications. For instance, metabolic engineering of the steroid-eater strains has allowed to obtain intermediaries of industrial value. However, there are still some drawbacks that must be overcome, such as the redundancy of the steroid catabolism genes in the genome and a better knowledge of its genetic regulation. KshABs and KstDs are key enzymes involved in the aerobic breakage of the steroid nucleus. *Rhodococcus ruber* Chol-4 contains three *kshAs* genes, a single *kshB* gene and three *kstDs* genes within its genome. In the present work, the growth of *R. ruber ΔkshA* strains was evaluated on different steroids substrates; the promoter regions of these genes were analyzed; and their expression was followed by qRT-PCR in both wild type and *ksh* mutants. Additionally, the transcription level of the *kstDs* genes was studied in the *ksh* mutants. The results show that KshA2B and KshA1B are involved in AD metabolism, while KshA3B and KshA1B contribute to the cholesterol metabolism in *R. ruber*. In the *kshA* single mutants, expression of the remaining *kshA* and *kstD* genes is re-organized to survive on the steroid substrate. These data give insight into the fine regulation of steroid genes when several isoforms are present.

## 1. Introduction

Sterols and related steroids are widespread in nature due to their important physiological activities that vary from their structural role in the eukaryotic membrane to their action as hormones. These molecules are highly resistant to degradation due to their hydrophobic nature and reduced presence of functional groups; nevertheless, it is possible to find bacteria that can degrade steroid compounds [1]. Several species of actinobacteria, including *Mycobacterium* and *Rhodococcus*, are able to grow on steroids such as cholesterol (CHO), phytosterols, androgens, bile acids or estrogens as sole carbon and energy source [2,3,4,5,6]. Although different pathways for aerobic or anaerobic steroid degradation may co-exist in the same organism, the role and participation of steroid genes involved in cholesterol catabolism have been clarified by biochemical and genetic studies [1,3,4,5,7,8,9,10]. A generalized microbial steroid 9,10-*seco* catabolic pathway scheme is depicted in Figure 1. The genome search for catabolic steroid genes encoding clusters has resulted in important advances in metabolic engineering and sterol bioconversion of both *Rhodococcus* and *Mycobacterium* [3,11,12,13,14,15]. These findings have allowed the production of valuable steroid intermediates such as 9α-hydroxyandrost-4-ene-3,17-dione (9OH-AD), 4-androstene-3,17-dione (AD), 1,4-androstadiene-3,17-dione (ADD) or 22-hydroxy-23,24-bisnorchol-1,4-dien-3-one (1,4-HBC) [16,17,18,19]. Some of these steroids or their derivatives are used for medical purposes in humans, livestock and aquaculture [20].

On the other hand, the research on steroid catabolism is also a challenge for solving environmental bioremediation such as the steroid liberation to the environment from different sources (e.g., the presence of estrogens, a type of endocrine disruptor and group 1 carcinogens that endanger life, in sewages waters) [7]. In fact, both biogenic (natural) and anthropogenic steroid hormones are frequently detected in soils and aquatic environments, in which, even at low concentrations, their release has adverse physiological effects on aquatic organisms [7,20]. Actinobacteria are reported to be active estrogen degraders and therefore, this microbial degradation is crucial for removing steroid derivatives from polluted ecosystems [7,20,21]. Lastly, the steroid catabolic pathway is also important for understanding the pathogenicity and virulence of pathogenic actinobacteria as for their survival they rely on host cholesterol degradation [22,23].

To develop new and better biotechnological applications to work in all these fields, a regulation of all these processes should be unraveled. For instance, the actinobacteria steroid catabolism is regulated through two HTH-type transcriptional regulators, KstR and KstR2 [24,25]. The consensus motif recognized by the two TetR-type transcriptional regulators is known; KstR controls nearly 100 catabolic genes and binds as a dimer to the consensus motif TNNAACGTGTTNNA [25,26] and KstR2 controls the expression of at least 15 genes involved in the lower catabolic pathway [24]. This regulator specifically binds to an operator region of 29 nucleotides containing the palindromic sequence AAGCAAGNNCTTGCTT [24,27]. However, many genes for steroid catabolism have yet to be identified, and many of the promoter and regulation issues concerning this degradative pathway are poorly characterized and remain to be fully elucidated. Key enzymes in the general scheme of the bacterial aerobic steroid catabolism are the 3-ketosteroid 9α-hydroxylase (KshAB: androsta-1,4-diene-3,17-dione, NADH: oxygen oxidoreductase (9α-hydroxylating); EC 1.14.13.142) in combination with a 3-ketosteroid-Δ^1^-dehydrogenase activity (KstD) (see Figure 1). Both are responsible for the steroid nucleus (rings A/B) breakage. KshAB initiates the opening of the steroid ring by the 9α-hydroxylation of the C9 carbon of 4-ene-3-oxosteroids (e.g., AD) or 1,4-diene-3-oxosteroids (e.g., ADD), transforming them into 9OH-AD or 9α-hydroxy-1,4-androstadiene-3,17-dione (9OH-ADD), respectively. On the other hand, the flavoprotein KstD converts 4-ene-3-oxosteroids (e.g., AD) to 1,4-diene-3-oxosteroids (e.g., ADD) by trans-axial elimination of the C-1(α) and C-2(β) hydrogen atoms [4].

*Rhodococcus* genus is characterized for their metabolic diversity partly due to their highly redundant biosynthetic pathways [28,29]. This is the case of *Rhodococcus ruber* strain Chol-4 (CECT 7469; DSMZ 45280), a strain isolated from a sewage sludge sample [30]. For instance, up to three homologous *kshA* genes and a single *kshB* gene have been described in different genomic regions of this strain [31]. The studies done with a series of *R. ruber* mutants (Δ*kshB* and single, double and triple Δ*kshA* mutants) have probed that KshA2 isoform is needed for the degradation of steroid substrates with short side chain; KshA3 works on those molecules with longer side chains and KshA1 acts as a more versatile enzyme related to the cholic acid (ACHO) catabolism and collaborating with KshA2 or KshA3 activities [31]. Although these enzymes are important within the steroid catabolism, there are not many studies done on genetic regulation of these activities, just for KstD promoters [32], a partial study on Ksh2 in *Rhodococcus erythropolis* strain SQ1 [33] and general studies on *Rhodococcus* gene promoters [34,35]. The findings of how the steroid clusters are regulated could be useful in metabolic engineering of *Rhodococcus* strains. In this line, to provide further insights into the role of cholesterol catabolism regulation, we studied the promoter regions of *ksh* and the expression of *ksh* and *kstD* genes in *R. ruber*.

## 2. Materials and Methods

### 2.1. Bacterial Strains, Plasmids and Growth Conditions

*Rhodococcus ruber* strain Chol-4 (CECT 7469; DSM 45280) was isolated from a sewage sludge sample [30]. This strain and the derived mutants ∆*kshA1*, ∆*kshA2*, ∆*kshA3* and ∆*kshB* [31] were routinely grown on Luria–Bertani (LB) or minimal medium (M457 of the DSMZ, Braunschweig, Germany) containing the desired carbon and energy source under aerobic conditions at 30 °C in a rotary shaker (250 rpm) for 1–3 days. For the steroid growth experiments, a LB pre-grown culture was washed two times with minimal medium prior to inoculation. Biological replicates (from 3 to 6) were performed for all growth experiments. CHO and AD were previously dissolved in methyl-β-cyclodextrin (CD) and prepared as described [36], whereas cholic acid (ACHO) was prepared in water. For the promoter growth experiments, cells were plated in LB agar with appropriate antibiotics and incubated at 30 °C for 3 days. All reagents for the cultures including antibiotics were purchased at Sigma-Aldrich, Spain. *Escherichia coli* strains were grown on LB broth at 37 °C, 250 rpm. Plasmids and bacterial strains used are listed in Table 1.

### 2.2. In Silico Analyses

DNASTAR Lasergene programs (DNASTAR, Inc., Madison, WI, USA) were used to analyze sequences and to design primers. The *R. ruber* strain Chol-4 genomic DNA has been previously sequenced [37]. The BioEdit program (Ibis Biosciences, Carlsbad, CA, USA) was used to perform local-blast alignments within the genome data (NCIB::ANGC00000000.2). Sequence promoters were analyzed using Promoter Hunter tool from PhiSITE [38], a tool for promoter search in prokaryotic genomes, using default parameters. The final score obtained for each PhiSITE sequence, −10 and −35, is an indicative of probability to function as a promoter.

### 2.3. Promoter Cloning and Characterization

All the putative promoters were cloned into pNVS-A, a plasmid-based promoter-probe vector constructed for the qualitative analysis of *Rhodococcus* promoters that carries the apramycin resistance gene as a genetic marker (*Am^R^*) [32].

The putative *ksh* promoter sequences were amplified by PCR. Amplicons covered the intergenic region from the end of the upstream flanking gene to the end of the first six amino acids codifying sequence of the *ksh* ORF. If needed, shorter versions of the intergenic region were also PCR amplified. The *Eco*RI-*Nru*I flanked promoter regions (*kshA1p*, *kshA2p*, *kshA3p*, *kshBp*, *kshA1pm*, *kshA2pm*, *kshA3pm*, *kshA3pm2* and *kshBpm*) were cloned into pNVSP1-A replacing the *kstD1* promoter. The recombinant vectors were designated as pNVSPA1-A, pNVSPA2-A, pNVSPA3-A, pNVSPB-A, pNVSPA1^m^-A, pNVSPA2^m^-A, pNVSPA3^m^-A, pNVSPA3^m2^-A and pNVSPB^m^-A, respectively. These vectors were digested with *Nru*I-*Hin*dIII to obtain the different constructions without the *Am^R^* gene. As control, the promoter-less plasmid carrying the *Am^R^* gene (pNVS-A) was used.

All the plasmids were checked by sequencing (Eurofins Genomics). Primers and vectors used in this work are listed in Table 1 and Table 2. PCR was performed under standard conditions using High Fidelity PCR PrimeStar (Takara, Japan) with glycerol 5% (*v*/*v*). T4 DNA ligase and PCR reagents were supplied by Sigma-Aldrich (Spain). Restriction enzymes were purchased from Takara, Japan. All DNA manipulations were performed according to standard molecular cloning procedures or following manufacturers’ instructions (NZYMiniprep and NZYGelpure from NZYtech, Portugal).

Each plasmid was introduced in *R. ruber* strain Chol-4 by electroporation (200 µL cells with 1 µg DNA at 400 Ω, 25 mA, 2.5 µF and 10–11 ms). Cells were then suspended in 800 µL of LB and kept for 6 min at 46 °C and, after that, kept for 6 h at 30 °C without shaking. Finally, they were plated on LB Agar with 200 μg/mL kanamycin and kept at 30 °C. Different colonies of *R. ruber* strain Chol-4 harboring the recombinant plasmids were picked up and grown on LB Agar at 30 °C. When needed, apramycin at 300 μg/mL and/or kanamycin at 200 μg/mL was added to the medium.

### 2.4. PCR and RT-qPCR Analysis

RNA samples for RT-qPCR experiments were obtained from mid-log exponential phase cultures (OD_600nm_ 0.7–0.8). Total RNA was prepared as previously stated [32].

To quantify the expression of the three *kstDs*, three *kshAs* and the *kshB* gene of *R. ruber*, a RT-qPCR analysis was performed using RNA from strains cultured in M457 minimal medium containing the desired carbon and energy source. Control cells were grown at 25 mM sodium acetate carbon source while sample cells were grown at 10 mM AD, 10 mM CHO or 10 mM ACHO carbon source. The RNA quality was assessed by using Bioanalyzer 2100 (Agilent, Barcelona, Spain). cDNA was synthetized using 1 μg of RNA with the High Capacity RNA to cDNA Kit (Applied Biosystems, Madrid, Spain). The RT-qPCR analysis of cDNA was performed on Applied Biosystems QuantStudio 12K Flex Real-Time PCR Systems. The reaction conditions were 10 min at 95 °C followed by 35 cycles of 15 s at 95 °C and 1 min at 60 °C for extension. The temperature of the melting curve was from 60 to 95 °C. The *16S* gene was used as an internal control to normalize messenger RNA levels [39]. All reactions were performed in duplicate in at least two biological replicates at the Genomic Unit of Universidad Complutense de Madrid. Relative expression level was calculated using the 2^−ΔΔCT^ method [40] indicating the expression level detected for each gene with respect to its own level on sodium acetate. Data were represented using GraphPad Software (San Diego, CA, USA).

## 3. Results and Discussion

### 3.1. kshs Promoters Search

To study the *R. ruber ksh* promoters, we analyzed the intergenic region between the four *ksh R. ruber* genes and their upstream flanking gene, as depicted in Figure 2. First, we searched for KstR/KstR2 consensus binding sites within these intergenic regions. A double consensus KstR motif was found upstream of *kshA3* and a single motif laid upstream of the *kshB Rhodococcus* ORF. No KstR2 motifs were found in any of the studied regions (Figure 2).

Subsequently, we searched for ribosomal binding site (RBS) sequences in the *ksh* intergenic regions. It is reported that some steroid genes such as *kstD3* ORF are leaderless and do not display an RBS [32], and, therefore, we analyzed if this was also the case in the *kshA* promoters. As reference, we used the synthetic RBS AAGGAGG that has been proved to be effective in the Gram-positive *Streptomyces* [41] and also the strongest RBS found for effective expression of genes in actinobacteria: (AA)GGA(G)G(AA)A(AAA)CAT-ATG [42]. In the analyzed *ksh Rhodococcus* intergenic fragments, this RBS was partially found in all cases at different distances from the initial codon (Figure 2 and Appendix A). The analysis in silico of these intergenic regions by PhiSITE yielded putative cis-regulatory signals -35 and -10 for prokaryotic promoter that are depicted in Figure 2. In parallel, as the absence of a -35 motif seems to be a characteristic of actinobacterial promoters [43,44], we also searched for -10 regions similar to those described for *kstDs Rhodococcus* promoters [32] and mycobacterial promoters [43]. It must be noted that one of these promoters, the *Rhodococcus kstD1p* (TAGTGT), is identical to the -10 T101 promoter described for mycobacteria [43].

The *kshA1* intergenic region displays a putative promoter given by PhiSITE which forms a palindromic sequence. This region also contains two other putative -10 boxes in the surroundings of the PhiSITE promoter. The first one is similar to both *kstD1p* (1p) and *kstD2p* (2p)-10 motifs (TAGCGT vs. TAGTGT and TAGCCT, respectively) (Appendix A and Figure 2). The second one, TTTCCT, is not similar to any of the mycobacterial promoters described but it resembles to the *kstD2p* promoter.

The PhiSITE tool suggested two putative promoter regions for *kshA2* intergenic region (Figure 2). This intergenic region also contains two other putative -10 regions (TGACCT and ATGCCT) that resemble *kstD2p* but not any mycobacterial promoter (Appendix A).

The *kshA3* intergenic region contains the motif TTGTGT similar to the *kstD1p* (TAGTGT) lying between the two KstR motifs, while there are no sequences that resemble *kstD2p* or *kstD3p* promoters. There are also two PhiSITE putative promoters with similar score, being one coincident with the first KstR motif and the other just before the appearance of the second KstR motif (Figure 2).

On the other hand, the *kshB* intergenic region presents many putative promoter motifs that resemble *kstDp* and mycobacterial promoters, all of them lying near but upstream of the KstR binding site (Appendix A and Figure 2) The PhiSITE putative promoter with higher score is also coincident with the KstR motif.

### 3.2. Promoter Cloning and Characterization

To characterize the functionality of the promoter regions of the three *R. ruber kshAs* and *kshB*, the promoter-test vector pNVS-A suitable for this strain [32] was employed. *R. ruber* is sensitive to apramycin, and therefore we chose the expression of a gene encoding this resistance as a proof of promoter activity.

The four *ksh* intergenic regions along with the first 21 bases of each *ksh* were PCR amplified and transcriptionally fused to the apramycin resistance gene. The vector pNVS-A carrying the apramycin resistance gene (*Am^R^*) but without any upstream promoter region was used as a negative control. As a second control, a set of pNVSP vectors that contained the promoter region but do not have the apramycin resistance gene was also used. After electroporation of *R. ruber* with the recombinant plasmids, only the cells harboring the plasmids with a putative promoter and the apramycin resistance gene were able to grow on apramycin and kanamycin, while cells harboring the pNVS-A or the pNVSPs vectors were only able to grow on kanamycin plates (Figure 3A). Therefore, under the conditions used, the *R. ruber* intergenic sequences cloned in pNVS-A contained functionally active promoters.

To define a minimal functional promoter region, the apramycin promoter-test was performed once more but reducing the intergenic region (Figure 2). New primers were designed, hence only the putative promoter identified by PhiSITE nearest to the ORF was included in the amplicons. In this case, *R. ruber* cells harboring plasmids pNVSPA1^m^-A, pNVSPA2^m^-A and pNVSPB^m^-A were able to grow on either kanamycin or apramycin while their controls promoter-less or apramycin-less did not (Figure 3B). However, the promoter pNVSPA3^m^-A construction that included the region amplified by CH585 and CH433 for *kshA3* was not functional. Therefore, a second region promotor was designed for this ORF from CH593 (Figure 2). The recombinant vector pNVSPA3^m2^-A was transformed in *R. ruber* and in this case was able to express the apramycin resistance (Figure 3B). As the first proposal of a minimal promoter for *kshA3* ORF was not able to carry out the expression of the apramycin gene, it was concluded that any of the motifs found including both KstR boxes near the upstream flanking gene cannot be discarded for a functional promoter region (Figure 3B).

Recently, a review on the advances in genetic toolkits and methods for engineering *Rhodococcus* has been published, and, in parallel, a set of *Rhodococcus* promoter-RBS combinations with fine-tuning of different activity levels have been built and compiled [35,45]. The *Rhodococcus* promoter regions described here can contribute to enrich the set of genetic parts for gene expression control in this strain.

### 3.3. Growth of R. ruber Strain Chol-4 ΔkshA Strains in Different Steroids Substrates

Growth data in *R. ruber* Δ*kshA* double mutants using different steroidal compounds, such as AD, CHO and bile acids (ACHO) as sole carbon and energy source, gave us insight into the possible physiological roles of each *kshA* [31]. Two different KshA isoforms could contribute to the growth in AD (KshA1 and KshA2) and CHO (KshA3 and KshA1), while only KshA1 was involved in ACHO. To go further into the role of each KshA isoform and how growth on steroids is affected when only one of the three *kshA* is deleted, growth experiments of single *ksh R. ruber* mutants and wild type (WT) were performed by using AD or CHO as sole carbon and energy source. Sodium acetate was used as control substrate. The maximum absorbance reached at 600 nm and the doubling time are shown in Table 3 and Appendix A.

As expected, growth rate on sodium acetate was similar in all the *R. ruber* mutants and WT. This similar behavior allowed us to use sodium acetate as a suitable substrate control later in the transcription experiments (see Section 3.4).

As shown in Table 3, mutants that keep the KshA2 functional (that is, WT, Δ*kshA1* and Δ*kshA3*) displayed a similar growth trend in AD, while the Δ*kshA2* strain that keeps only *kshA1* and *kshA3* ORF increased the doubling time from 2.2. (WT) to 21.44 h. These data are similar to the ones reported for the Δ*ksh2,3* double mutant which only contained KshA1 activity that also yielded a doubling time of 21.9 h. In contrast, the mutant that only displays KshA3 activity (Δ*kshA1,2* double mutant) was not capable of growing on this substrate [31]. This result confirms that KshA2 is primarily involved in AD metabolism followed by KshA1, while KshA3 activity does not contribute to the AD metabolism in *R. ruber*.

On the other hand, all the single *ksh* mutants present similar growth parameters on cholesterol as the WT. From previous data reported, we know that KshA2 does not intervene in the CHO metabolism. Actually, in the double mutant Δ*kshA1,3* (only KshA2 activity acting), the growth on CHO for Δ*kshA1,3* was minimal (Abs_600nm_ = 0.3 and a doubling time of 15.1 h), and it seemed to be due to a residual growth for the cholesterol side-chain non-Ksh-mediated degradation [31]. Therefore, CHO growth depends on KshA1 or KshA3. The coexistence in the cell of both KshA1 and KshA3 does not yield better growth parameters than when they were alone or with KshA2. Accordingly, we can conclude that the absence of KshA3 is compensated by KshA1 and vice versa, and both isoforms contribute equally to the cholesterol metabolism in *R. ruber*, while KshA2 is not necessary in cholesterol metabolism in *R. ruber*. This relationship between isoforms could be more complex if the number of variants increases within the cell. A higher isoform multiplicity on KshAs than the three ones of *R. ruber* has been reported in some *Rhodococcus* such as. *R. rhodochrous* and *R. jostii* RHA1 which contain up to five different KshAs within their genomes. The phylogenetic studies of the *Rhodococcus* KshAs isoforms have shown that enzymes that belong to the same branch share similar functions and act with a similar spectrum of substrates [31,46,47]. For instance, KshA1 from both *R. ruber* and *R. rhodochrous* and KshA3 from *R. jostii* RHA1 prefer substrates with a carboxylate side chain at C17 and are involved in cholate catabolism. KshA2 orthologs are involved in the degradation of steroids with none or a short side chain (e.g., AD, ADD, testosterone and dehydroepiandrosterone). KshA3 and similar isoforms belong to the cholesterol degradation branch [31,46,47,48,49]. The other two KshAs in *R. rhodochrous* are KshA4_DSM43269_ that is similar to KshA2_DSM43269_ and KshA5_DSM43269_ that hydroxylates a broader range of steroid substrates [46,47]. It would be interesting and useful to identify and broaden the range of substrates for each isoform; this could show how they cooperate during the degradation of different steroidal compounds.

### 3.4. Transcriptional Analysis of kshA Genes in R. ruber Strain Chol-4

Currently, there is not a quantitative transcriptional study of the *ksh* genes in *Rhodococcus* genus. In the present work, we studied the transcription level of the three *kshA* genes and the *kshB* gene by RT-qPCR in the WT strain. Further, we analyzed the expression of the three *ksh* genes of *R. ruber* in the *kshA* mutant strains. Since KstDs are also involved in the steroid nucleus (rings A/B) breakage (Figure 1), we studied the expression of these genes.

#### 3.4.1. Transcriptional Studies in the WT Strain

RNA samples were prepared from cultures grown on M457 mineral medium supplemented with sodium acetate (used as substrate control), AD and CHO as source of energy and carbon. WT was additionally grown on ACHO. Values obtained for the expression of the *kshA* or *kshB* genes in sodium acetate in all the strains tested were considered as the baseline expression with a value of 1. We found that, in WT strain, genes *kshA* and *kshB* were constitutively transcribed under all the conditions tested in our assays at a basal level. However, in the presence of CHO, AD or ACHO, they were specifically upregulated. In fact, WT cells growing on AD showed a 22,438-fold increase of *kshA2* transcription response, the highest induction of all genes and conditions tested, while *kshA3* almost did not present any fold change (1.12) and *kshA1* showed a slight (2.34-fold) change (Figure 4). These data confirm that the presence of the KshA2 isoenzyme is strongly reinforced in *R. ruber* AD catabolism while KshA1 is mildly induced.

On the other hand, *kshA3* was the most upregulated gene with a 363.25-fold change in cholesterol, while *kshA1* and *kshA2* induced 6.48- and 6.95-fold change, respectively. *kshB* gene (the second component of the Ksh activity) was constitutively transcribed since it was present in all conditions tested, even sodium acetate, but it was also upregulated under CHO with a 3.2-fold change.

The high induction for *kshA3* in CHO is not surprising because, as discussed above (Section 3.1), there is a KstR binding motif in the promoter region of *kshA3* (Figure 2). KstR and KstR2 are highly conserved TetR family transcriptional repressors that regulates more than 70 genes but not all these genes are involved in the cholesterol catabolism [24,25]. All these genes are de-repressed by growth on cholesterol due to the generation of an early derivative CoA thioester cholesterol metabolites that interacts with KstR, the 3-hydroxy-cholest-5-en-26-oyl-CoA [50]. Therefore, the higher transcription in CHO for *kshA3* is according to this fact. *kshB* promoter also contains a KstR motif (Figure 2) that, for similar reasons, can explain why it is more induced on CHO. KshA1 can substitute KshA3 in the CHO metabolism as the presence of one of them is enough to reach a similar growth on CHO in *R. ruber* (Table 3), and, as expected, it is also induced on CHO (Figure 4). Surprisingly, KshA2 that does not contribute to CHO metabolism, showing a similar induction level to KshA1 (around six times more). This could be explained by the presence of basal levels of some steroid enzymes, which might allow multiple steroid derivatives to be generated as soon as sterol is present. Some of these subproducts might activate the expression of *kshA2*. The induction of this gene could help the cell to keep the metabolic flux throughout the pathway.

Lastly, *kshA1* and *kshA2* transcription levels increased 8.8- and 18.3-fold, respectively, while *kshA3* mRNA levels were not affected when growing on cholic acid (Figure 4). KshA1 is reported to be involved in the cholic acid catabolism in *R. ruber* [31] and therefore the increase in their mRNA levels is consistent. The induction of *kshA2* on this substrate was however unexpected, and, once more, KshA2 could be induced as a reinforcement for its role in some intermediaries of the cholic catabolism.

Additionally, the results show that *kshA3* and *kshB* were only induced in CHO while transcription levels of *kshA1* and *kshA2* were increased under all steroid conditions. We suggest that for *kshA1* and *kshA2* there must be other steroid regulators involved apart from KstR and KstR2 as other authors have previously suggested [51]. Recently, a motif for LuxR repressor for steroid genes such as 3,17β-HSD in *Comamonas testosteroni* ATCC11996 has been characterized [52]. LuxR binds to palindrome DNA sequences such as TTCCTCCGCGTTCGCGGAGGAACGAA blocking the transcription start point until testosterone is present in the medium. Precisely, we can find a palindrome sequence near the putative promoters found for *kshA2* and *kshA1* ORFs (Figure 2). We propose that other steroid regulators than KstRs must be acting repressing these steroidal promoters. If identified ligands for KstRs are 3-oxocholest-4-en-26-oyl-CoA and 3-oxo-23,24-bisnorchol-4-en-22-oyl-CoA and are described to occur early in cholesterol catabolism [50], the inductors of *kshA1* and *kshA2* promoters could belong to different stages of the catabolism (upper and intermediate) as the major induction for kshA2 promoter was obtained when growing on AD. The kind of repressor of these promoters is subject of further studies beyond this work.

The induction of some of the isoforms of *kshA* by CHO or AD has also been reported by conventional RT-PCR in other *Rhodococcus* such as in *R. erythropolis* or *R. rhodochrous* [33,46]. The five *kshA* homologs (*kshA1*_DSM43269_ to *kshA5*_DSM43269_) identified in *R. rhodochrous* DSM43269 showed a different expression pattern [46]. *kshA1*_DSM43269_ was induced in ACHO cultures and faintly in CHO, while its homolog *kshA1* in *R. ruber*, although had a similar expression pattern with ACHO (8.79-fold change), it was also induced in the presence of CHO or AD (6.48- or 2.34-fold change, respectively). *kshA2*_DSM43269_ was induced on AD and ACHO cultures but this isoenzyme does not share a defined identity with any of the three *kshAs* of *R. ruber*. *kshA2* of *R. ruber* share some common characteristics with *kshA2*_DSM43269_, *kshA4*_DSM43269_ or *kshA5*_DSM43269_, which were the highest expressed genes in *R. rhodochrous* AD cultures. On the other hand, although *kshA3* genes of *R. ruber* and *R. rhodochrous* displays a high identity between them, their expression patterns are not the same. In fact, *kshA3*_DSM43269_ gene was induced by AD [46] while *R. ruber* KshA3 was not induced on this substrate (1.12-fold change) but on CHO (363-fold change). When compared the upstream regions of these genes (Appendix A), we found a strong similitude with *kshA3s* from both *R. ruber* and *R. rhodochrous* keeping the two KstR motifs in both strains. *kshA1* upstream regions display a partial similitude while there is no correspondence in the case of *kshA2* (Appendix A). All these differences, despite the structural homology between genes, reveal a fine tuning in the involvement of each KshA enzyme that seems to depend on the strain.

#### 3.4.2. Transcriptional Studies in the Mutant Strains

When one KshA isoform is absent, an expected arrangement of the rest of the isoforms could be a logical assumption. To confirm this hypothesis, we have also studied the expression levels of the three *kshAs* and *kstDs* genes in the single mutants’ ∆*kshA1*, ∆*kshA2* and ∆*kshA3* strains grown on different steroids (Figure 5). In ∆*kshA1* and ∆*kshA3* strain the expression of *kshA2* gene was highly induced on AD substrate similarly to the observed values in the WT (Figure 4). These data are coherent, as we know from growth data that the isoform KshA2B is the predominant one on this substrate (Figure 6). However, what happens when KshA2 is not active? The ∆*kshA2* strain showed a higher induction of *kshA1* with respect to the WT induction (7.29- and 2.34-fold change, respectively, on AD growth). Taking into account that the general levels of *kshA1* expression are rather low, a difference of 3.1-fold change is high. These results confirm he biological characterization of KshA1 previously published [31] and the growth data obtained in this work in which KshA1 was able to replace the activity of KshA2, the main isoenzyme for AD catabolism. Consequently, KshA1B could be the predominant form in the absence of KshA2 isoform (Figure 6). Regarding *kshA3*, there was also a slight upregulation with a 1.76-fold change compared to the 1.12-fold change of the WT.

KshA3 and KshA1 contribute both to CHO catabolism in *R. ruber*. The high induction of *kshA3* of two orders of magnitude with respect to the *kshA1* induction obtained in the WT strain, KstR-dependent, makes us propose that KshA3B is the mean isoform on this growth (Figure 6). In the ∆*kshA2* strain, both isoforms, KshA1 and KshA3, obtained higher induction rates than the WT on CHO growth (11.3 versus 6.48 and 535 versus 363, respectively). In the ∆*kshA1* strain, *kshA3* is over expressed (a 479-fold change) following the same pattern than the WT (a 363-fold change) (Figure 5). This difference could compensate the lack of *kshA1*, which, despite its secondary role, seems to have a reinforcement function. On the other hand, in the case of the ∆*kshA3* mutant the expression profile shows an increase for *kshA1* expression (from 6.48 in the WT to 7.8 in the mutant). This increase seems to be enough for performing the CHO catabolism as there is no difference on the growth data with respect to the WT that keeps both *kshA* genes (Table 3). This suggests that the role of KshA1 and KshA3 can be interchangeable on the CHO catabolism and that the lack of one of them does not suppose a limiting stage in *R. ruber*.

Both Kshs and KstDs are involved in key steps of the steroid catabolism [7]. For this reason, we also evaluated if *kstD* redundancy in *R. ruber* (KstD1, KstD2 and KstD3) suffer changes in their induction level when a particular *kshA* is missing. Data show (Figure 5) that, on AD, *kstD1* was upregulated in all *kshAs* mutants (99-fold, 92-fold and 41-fold change in ∆*kshA1*, ∆*kshA2* and ∆*kshA3*, respectively) with respect to the WT (13.6-fold change, [32]). KstD1 acts on 9OH-AD, an intermediary generated by the action of KshAB. These results show that the lack of one of the two KshAs involved on AD degradation (KshA1 and KshA2) is enough to upregulate *kstD1*. Surprisingly, even when KshA1 and KshA2 are present and KshA3 is missing (∆*kshA3* strain), an induction of *kstD1* was detected but in a lower degree (41-fold change in ∆*kshA3*). *kstD2* expression levels are also induced on AD (7.6 in ∆*kshA1*, 9.3 in ∆*kshA2* and 6.3-fold change in ∆*kshA3*) with respect to the 0.7-fold change in the WT. KstD2 is the main enzyme involve in AD catabolism in *R. ruber* Chol-4 [32,53], which transforms AD to ADD (Figure 1). The data might suggest that, in the absence of KshA2B, an upregulation of KstD2 and KstD1 could help to divert the catabolic pathway in order to prevent AD or other steroid intermediaries’ accumulation into the cell. A similar idea has been also suggested in *R. erythropolis.* in which the authors suggested that a regulation of KstD and Ksh activities would be advantageous to the cell to keep low the intracellular levels of ADD [33].

On the other hand, *kstD3* levels practically remained equal in the three mutants (2.4-, 1.6- and 0.8-fold change, respectively, compared to 0.6-fold change in the WT, Figure 5). This fact makes sense due to the KstD3 involvement in CHO rather than in AD catabolism.

Regarding the growth on CHO, relative expression levels of *kstD2* in the *kshA* mutants (Figure 5) were unaffected with respect to the WT values or even downregulated (0.3-, 2.6- and 1.3-fold change in the *kshA1*, *kshA2* and *kshA3* mutants, respectively, compared to 2-fold change in the WT). *kstD1* increases from 7.6-fold change in the WT and 8-fold change in *kshA1* mutant to 16- and 34-fold change in the *kshA3* and *kshA2* mutants. Finally, *kstD3* expression rises from 240–250 (WT and *kshA3* mutant) to 320- and 537-fold change in *kshA1* and *kshA2* strains. Taking into account that KstD3 and KstD2 are the main enzymes involved in CHO catabolism and that they display a different preference for steroidal substrates [32], an upregulation of these enzymes to compensate the system might be expected. However, only *kstD3* is upregulated while *kstD2* keeps almost the same level as the WT. On the other hand, KstD1, which does not have a particular role in CHO degradation, is induced in the absence of KshA1 or KshA2. KstD1 may have a reinforcement role in *R. ruber* steroid catabolism and therefore its induction could help in the degradation of secondary metabolites generated from the main catabolic pathway. On the contrary, *kstD2* levels are more reluctant to be modified in the upper CHO catabolism and more sensitive to middle steroid intermediaries.

The results suggest that there is an interaction between KshABs and KstDs as the lack of any KshA has an impact on the expression of most *kstDs* depending on the steroid. On AD, *kstD1* and *kstD2* are upregulated, while in the case of CHO the regulation is more complex.

There are other cases reported of cross-inducing steroid genes in the literature. For instance, *kshA1*_DSM43269_ is induced on cholesterol but has ADD and 1,4-BNC as preferred substrates [46]. Recently, it has been reported by transcriptome analysis that *kshA* C7H75_RS08645 (71% amino acid identity with *R. ruber* KshA1) and *kshA* C7H75_RS02785 (65–68% amino acid identity with *R. ruber* KshAs) expression were upregulated 11–14-fold in the estrogen-grown *R. equi* DSSKP-R-001 [6]. However, the biochemical mechanism for estrogen catabolism in *Rhodococcus* sp. strain B50 goes via the 4,5-*seco* pathway requiring the involvement of more 4-hydroxyestrone 4,5-dioxygenase activities than Kshs that is involved in the cholesterol/androgen catabolism via 9,10-*seco* pathway [21]. The relationship among KshAB and KstD isoforms requires a more careful further study and indicates that the absence of an activity in the cell forces readjusting more than one pathway. Our data also indicate that the roles found for each Ksh are not absolute and, in vivo, isoforms can be involved in supplementary roles such as KshA1 and KshA3 in the cholesterol metabolism or KshA2 and KshA1 in the AD metabolism. This versatility could not only represent an environmental advantage, but it also could help as suggested for a dynamic and fine-tuned steroid catabolism by preventing the accumulation of steroid intermediates [7]. This adjustment must not be only limited to steroid genes. The transcriptome analysis of *R. equi* DSSKP-R-001 also indicated that, in addition to the steroid-degrading enzymes, transporters and other metabolism related enzymes can also be important in the steroid catabolism process as they are differentially expressed [6]. Therefore, for a particular scenario, *Rhodococcus* strains will adjust their versatile gene expression for the best adaptation in each changeable situation.

## 4. Conclusions

The new data presented here contribute to a better understanding of the steroid catabolism and the transcriptional regulation mechanisms involved in *R. ruber*. The single presence of KshA3 or KshA1 activity in the cell is enough for CHO metabolism, while KshA2 is mainly involved in the AD metabolism, and its expression is highly induced on this substrate. When KshA2 is missing, KshA1 replaces its activity, although it takes longer to resume the growth on AD. *kshA3* and *kshB* belong to the KstR regulon, and, consequently, the induction of these genes is limited only to the presence of CHO. On the other hand, *kshA1* and *kshA2* promoter regions lack the KstR motif and therefore their promoters must be regulated in a different way; they are induced in the presence of AD and ACHO. The increase in expression levels of KshA2 on AD might have as a result a preferential binding of this subunit to KshB at the expense of KshA3 and KshA1, which should compete among them for this subunit. When KshA2 is missing, KshA1 levels increase, and this isoform is then preferably coupled to KshB to keep the cell growing on AD. Consequently, the growth and transcriptional results as a whole confirm that KshA2 and KshA3 are candidates to be the main enzymes to participate in the degradation of AD and CHO, respectively, while KshA1 is performing a supplementary role.

In this work, we also probed that mutations in a single *ksh* provoke adjustments in both *ksh* and *kstD* expressions of the other isoforms within the cell to compensate for the lack of that ORF. We conclude that these redundancies give the cell certain flexibility for growing on different conditions, being able to take the control over when one of the isoforms is not active. Due to the fact that the number of KshA isoenzymes depends on the *Rhodococcus* strain, the catabolism of steroids could have developed from a common core to a specific strain-dependent scenario. Further studies should be done to shed light on the fine tuning of these enzymes.

## Figures and Tables

**Figure 1 microorganisms-09-01171-f001:**
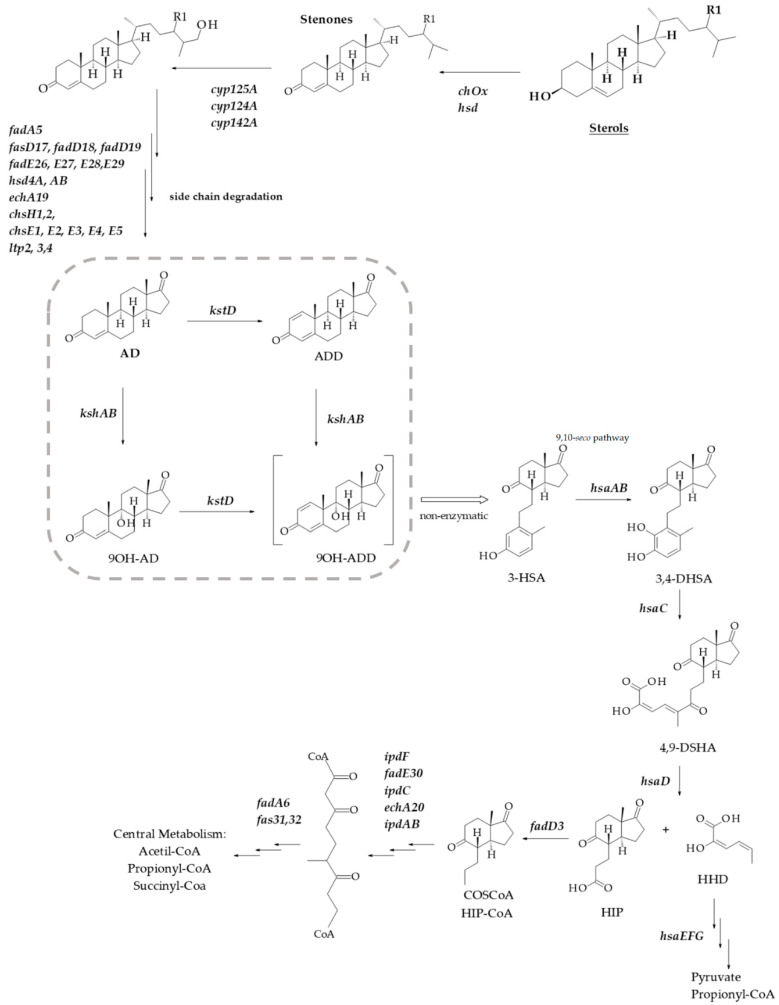
Scheme of aerobic sterol degradation in actinobacteria [4,5,7]. The pathway begins with the sterol’s oxidation to stenone by the action of a cholesterol oxidase (ChoX) or an hydroxysteroid dehydrogenase (HSD) The degradation continues with side chain cleavage via a mechanism similar to β-oxidation of fatty acids initiated by cytochromes P450s and in cycles of β-oxidation from the side chain. The steroid ring structure is degraded by oxygen-dependent opening and subsequent hydrolytic cleavage of rings A and B following the 9,10-*seco* pathway. In this central catabolic pathway, AD is transformed into 9OH-ADD with the involvement of KstD and KshAB activities (marked with a square). After that, the catabolism proceeds with the C and D rings cleavage. R1 indicates H, CH3 or C2H5 for various sterols. 3-HSA, 3-hydroxy-9,10-secoandrosta-1,3,5(10)-triene-9,17-dione; 3,4-DHSA, 3,4-dihydroxy-9,10- secoandrosta-1,3,5(10)-triene-9,17-dione; 4,9-DSHA, 4,5,9,10-diseco-3-hydroxy-5-9-17- trioxoandrosta-1(10),2-diene-4-oic acid; HHD, 2-hydroxy-2,4-hexadienoic acid; HIP, 3aα-H-4α(3′-propanoate)7a-β-methylhexahydro-1,5-indanedione.

**Figure 2 microorganisms-09-01171-f002:**
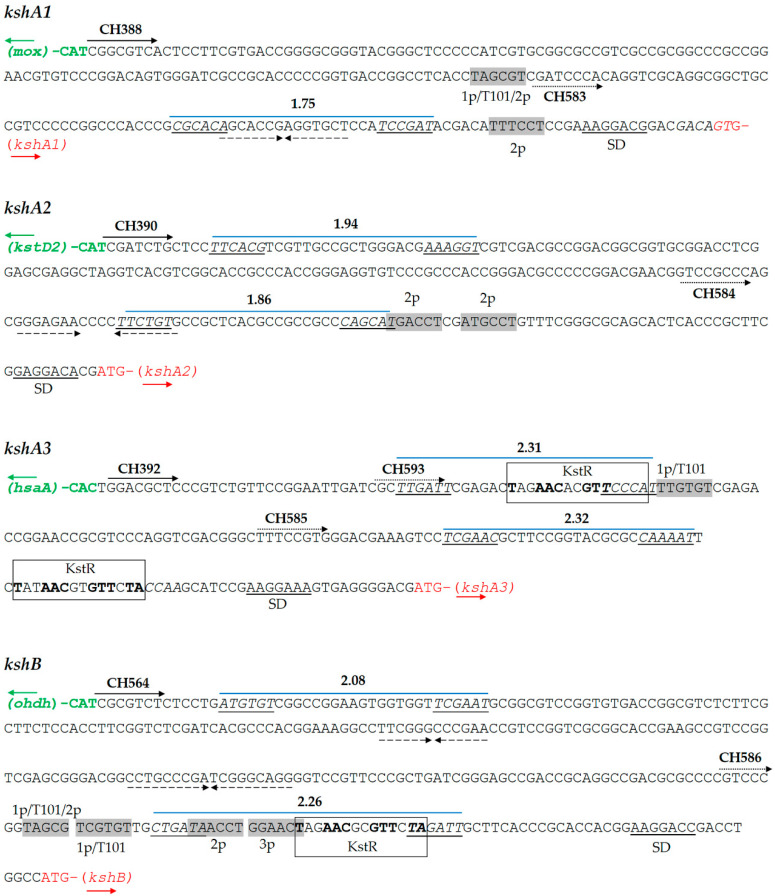
Analysis of promoter regions of *R. ruber ksh* genes. Predicted -35 and -10 boxes with Promoter Hunter tool with the final score obtained are shown in italics and underlined. Shine Dalgarno (SD) sequences are indicated. Primers used (CH) are shown in wide solid arrows (for the whole intergenic region) or in dotted arrows (reduced promoter region). Similar -10 promoter boxes to *kstD* or *Mycobacterium* promoters are shown in grey rectangles. Repetitive inverted sequences are marked in dashed arrows. KstR consensus binding motif appears in a rectangle. 1p: -10 *kstD1* promoter box. Abbreviations: T101, -10 *Mycobacterium tuberculosis* promoter box [43]; 2p, -10 *kstD2* promoter box; *mox*, ORF that codifies monooxygenase (KXF85873.1); *hsaA*, ORF that codifies a flavin-dependent mono-oxygenase that hydroxylates 3-hydroxy-9,10-seconandrost-1,3,5(10)-triene-9,17-dione (3-HSA) to a catechol; *odhd*, ORF that codifies an alcohol dehydrogenase (KXF85151.1). Other accession numbers: *kshA1* (D092_12170); *kshA2* (D092_19475); *kshA3* (D092_22970); *kshB* (D092_15920); and *kstD2* (D092_19480).

**Figure 3 microorganisms-09-01171-f003:**
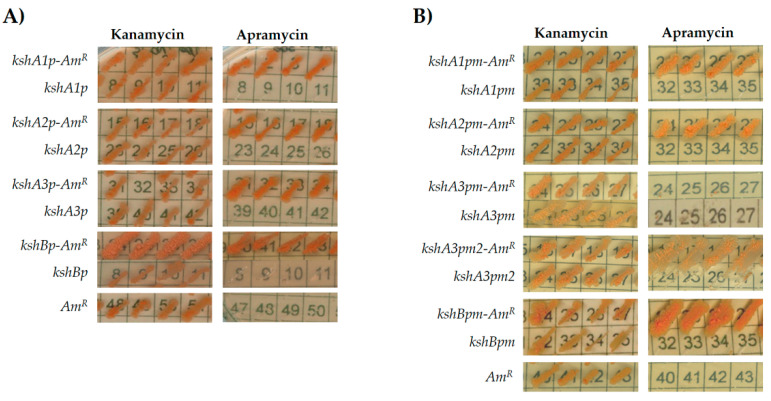
Comparative assessments of *R. ruber ksh* promoters: (**A**) intergenic regions; and (**B**) reduced promoter regions. Cells of *R. ruber* strain Chol-4 harboring different recombinant plasmids were grown on LB in the presence of either kanamycin (200 μg/mL) or apramycin (300 μg/mL). Abbreviations: *kshA/Bp*, pNVS vector containing the putative promoter region; *kshA/Bp-Am^R^*, pNVS vector containing the putative promoter region in frame to the apramycin gene; *kshA/Bpm*, pNVS vector containing the putative minimal promoters; *kshA/Bpm-Am^R^*, pNVS vector containing the putative minimal promoter coupled to the apramycin resistance gene; *Am^R^*: pNVS, vector containing the apramycin resistance gene without any promoter.

**Figure 4 microorganisms-09-01171-f004:**
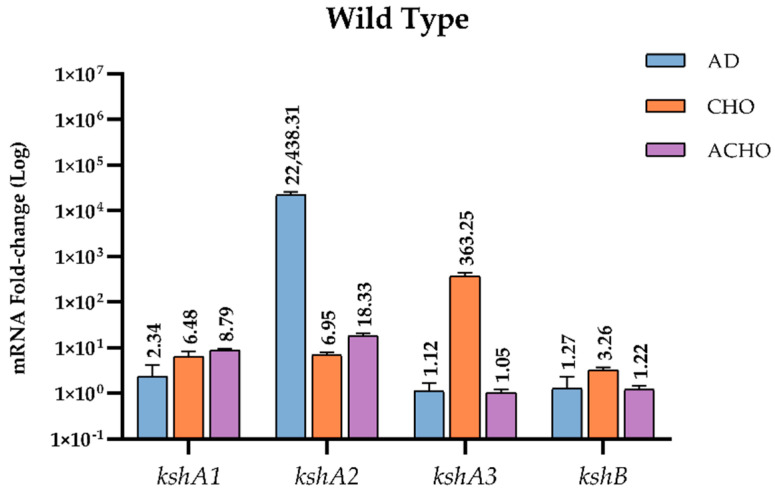
qRT-PCR analysis of *kshs* genes in *R. ruber* grown on different carbon sources. Gene expression was normalized relative to the expression of the 16S ribosomal subunit. Data obtained on sodium acetate were considered as the baseline expression and assigned as 1. Values are the means ± standard deviation (SD) (n = 2–3).

**Figure 5 microorganisms-09-01171-f005:**
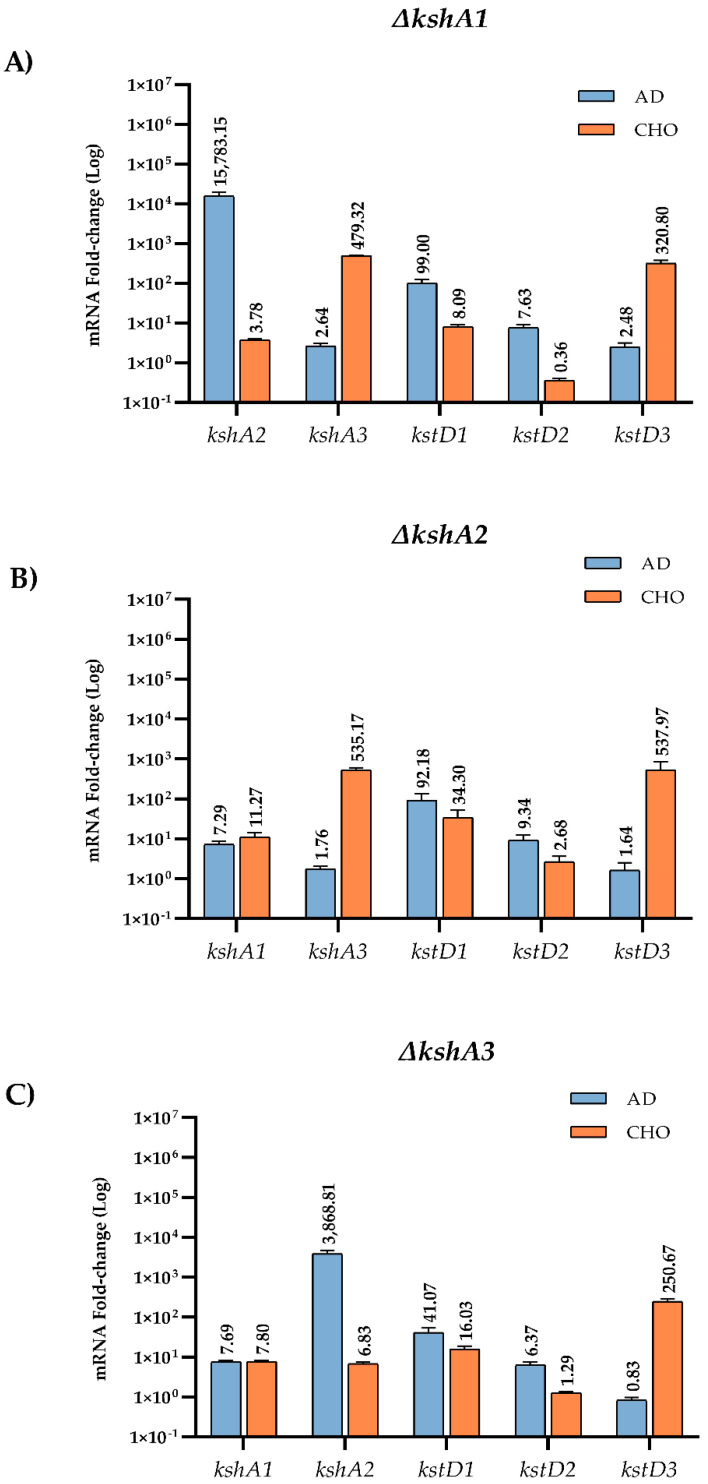
qRT-PCR analysis of *kshs* and *kstDs* grown on different carbon sources on *R. ruber ksh* mutants. Gene expression was normalized relative to the expression of the 16S ribosomal subunit. Data obtained on sodium acetate were considered as the baseline expression and assigned as 1. Values are the means ± standard deviation (SD) (n = 2–3). (**A**) *kshA1* mutant; (**B**) *kshA2* mutant; and (**C**) *kshA3* mutant.

**Figure 6 microorganisms-09-01171-f006:**
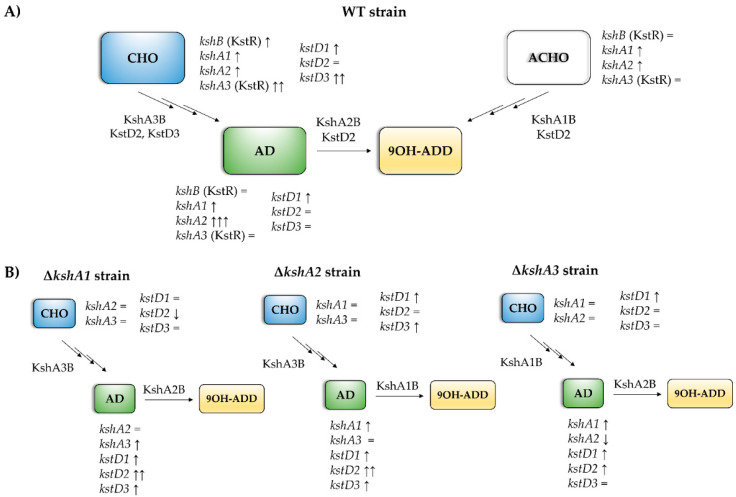
Scheme of the predominant *ksh* and *kstD* isoforms in the presence of different steroids. (**A**) Predominant isoforms in WT strain on different steroids. The main KshAB involved in different substrates is indicated. *ksh* genes with a strong dependence on KstR are shown in brackets. Transcription levels of each isoform are indicated related to the sodium acetate: =, less than 2-fold; ↑, more than 2-fold; ↑↑, more than 100-fold; ↑↑↑, more than 1000-fold. (**B**) Arrangements of isoforms transcription in the *ksh* mutants compared to WT. =, less than 2-fold; ↑, more than 2-fold; ↑↑, more than 10-fold; ↓, more than 2-fold (downregulation).

**Table 1 microorganisms-09-01171-t001:** Bacterial strains and plasmids used in this work.

Strain and Plasmids	Description	Reference
*E. coli* DH5α	*F’ endA1 hsdR17 (r_K_^−^ m_K_^+^) glnV44 thi-1 recA1 gyrA (Nal^R^) relA1 Δ(lacIZYA-argF) U169 deoR (φ80dlacΔ(lacZ)M15).*	Laboratory collection
*Rhodococcus ruber* strain Chol-4	Wild type phenotype, Nal^R^	[30]
Δ*kshA1 R. ruber*	Mutant in *kshA1* gene, Nal^R^.	[31]
Δ*kshA2 R. ruber*	Mutant in *kshA2* gene, Nal^R^.	[31]
Δ*kshA3 R. ruber*	Mutant in *kshA3* gene, Nal^R^.	[31]
Δ*kshB R. ruber*	Mutant in *kshB* gene, Nal^R^.	[31]
pGem-T Easy	Cloning vector *E. coli*, Ap^R^	Promega
pNVS	*Nocardia-E. coli* replicative shuttle vector pNV119 containing Terminator 1-mcs-terminator 0 from pSEVA351, Km^R^	[32]
pNVSP1-A	pNVS containing *kstD1* promoter region and the sequence of the first seven amino acids in frame to Apramycin resistance gen (*Am^R^*).	[32]
pNVSPA1	pNVS containing *kshA1* promoter region and the sequence of the first seven amino acids in the mcs. (*Eco*RI-*Nru*I, 0.285 kb)	This work
pNVSPA2	pNVS containing *kshA2* promoter region and the sequence of the first seven amino acids in the mcs. (*Eco*RI-*Nru*I, 0.298 kb)	This work
pNVSPA3	pNVS containing *kshA3* promoter region and the sequence of the first seven amino acids in the mcs. (*Eco*RI-*Sca*I, 0.244 kb)	This work
pNVSPB	pNVS containing *kshB* promoter region and the sequence of the first seven amino acids in the mcs (*Eco*RI-*Nru*I, 0.366 kb)	This work
pNVSPA1-A	pNVSPA1 in frame to *Am^R^* gene (*Nru*I-*Hin*dIII, 0.8 kb)	This work
pNVSPA2-A	pNVSPA2 in frame to *Am^R^* gene (*Nru*I-*Hin*dIII, 0.8 kb)	This work
pNVSPA3-A	pNVSPA3 in frame to *Am^R^* gene (*Nru*I-*Hin*dIII, 0.8 kb)	This work
pNVSPB-A	pNVSPB in frame to *Am^R^* gene (*Nru*I-*Hin*dIII, 0.8 kb)	This work
pNVS-A	A derivative of pNVSP1-A with the promoter P1 region deleted	[32]
pNVSPA1^m^	pNVS containing *kshA1* minimum promoter region and the sequence of the first seven amino acids in the mcs (*Eco*RI-*Nru*I, 0.155 kb)	This work
pNVSPA2^m^	pNVS containing *kshA2* minimum promoter region and the sequence of the first seven amino acids in the mcs (*Eco*RI-*Nru*I, 0.153 kb)	This work
pNVSPA3^m^	pNVS containing *kshA3* minimum promoter region and the sequence of the first seven amino acids in the mcs (*Eco*RI-*Nru*I, 0.142 kb)	This work
pNVSPA3^m2^	pNVS containing *kshA3* minimum promoter region and the sequence of the first seven amino acids in the mcs (*Eco*RI-*Nru*I, 0.213 kb)	This work
pNVSPB^m^	pNVS containing *kshB* minimum promoter region and the sequence of the first seven amino acids in the mcs (*Eco*RI-*Nru*I, 0.134 kb)	This work
pNVSPA1^m^-A	pNVSPA1^m^ in frame to *Am^R^* gene (*Nru*I-*Hin*dIII, 0.8 kb)	This work
pNVSPA2^m^-A	pNVSPA2^m^ in frame to *Am^R^* gene (*Nru*I-*Hin*dIII, 0.8 kb)	This work
pNVSPA3^m^-A	pNVSPA3^m^ in frame to *Am^R^* gene (*Nru*I-*Hin*dIII, 0.8 kb)	This work
pNVSPA3^m2^-A	pNVSPA3^m2^ in frame to *Am^R^* gene (*Nru*I-*Hin*dIII, 0.8 kb)	This work
pNVSPB^m^-A	pNVSPB^m^ in frame to *Am^R^* gene (*Nru*I-*Hin*dIII, 0.8 kb)	This work

**Table 2 microorganisms-09-01171-t002:** Primers and PCR conditions use in this work.

Primer	Sequence	PCR Conditions	Use
CH469 CH470	TACGAGGGCTACGACATC TACATCAGGACGAACTTGTT	Tm 60 °C, 1 min, 30 cycles	*kshA1* amplification in RT-qPCR, 52 bp
CH522 CH523	AAGACGTTCGAGCAGACAT GTTGTCGATCTTGGCCTTGT	Tm 60 °C, 1 min, 30 cycles	*kshA2* amplification in RT-qPCR, 50 bp
CH524 CH525	ACGCTGGTACGAGCAGTTCT GTCGACCTCGTACTCGAAGC	Tm 60 °C, 1 min, 30 cycles	*kshA3* amplification in RT-qPCR, 79 bp
CH581 CH582	ACCACGACTTCCTGCTGTTC GTAGACCAGCACGACCTTCC	Tm 60 °C, 1 min, 30 cycles	*kshB* amplification in RT-qPCR, 104 bp
CH505 CH506	GACATCGAGTTCACGGCCTA GGAGCCTTGCCGAAGTAGTC	Tm 60 °C, 1 min, 30 cycles	*kstD1* amplification in RT-qPCR, 50 bp
CH507 CH508	TCCTTCATCGTCGACCACAC CTGGCCGAACGACATGTAGT	Tm 60 °C, 1 min, 30 cycles	*kstD2* amplification in RT-qPCR, 69 bp
CH509 CH510	GCTACGACCACTACTACGGC CACCACCTTGATCGCGTAGA	Tm 60 °C, 1 min, 30 cycles	*kstD3* amplification in RT-qPCR, 92 bp
CH575 CH576	ATTAGTGGCGAAGGGTGAG CCCGAGGTCCTATCCGGTAT	Tm 60 °C, 1 min, 30 cycles	16S gen in RT-qPCR, 95 bp [39]
CH388	GGATCCTCTAGA**GAATTC**CGGCGTCACTCCTTCGTGACCGGG	Tm 56 °C, 30 seg, 30 cycles	Putative promoter of *kshA1* and its first 21 nucleotides of coding sequence with *Eco*RI/*Nru*I restriction sites, 285 bp
CH434	TTTAAACTGCAG**TCGCGA**GGACTTGCTGGAGCTCACTGTCGT		
CH583	GGATCCTCTAG**GAATTC**GATCCCACAGGTCGCAGGCG	Tm 55 °C, 30 seg, 30 cycles	Together with CH434, amplification of the putative minimum promoter region of *kshA1* and its first 21 nucleotides of coding sequence with *Eco*RI-*Nru*I sites, 155 bp
CH390	GGATCCTCTAGA**GAATTC**CGATCTGCTCCTTCACGTCGTTGCCG	Tm 56 °C, 30 seg, 30 cycles	Putative promoter of *kshA2* and its first 21 nucleotides of coding sequence with *Eco*RI/*Nru*I restriction sites, 298 bp
CH433	TTTAAACTGCAG**TCGCGA**GGGTGCGCTCTGCGTCATCGTGTC		
CH584	GGATCCTCTAGA**GAATTC**CGCCCAGCGGGAGAACCCCTT	Tm 60 °C, 30 seg, 30 cycles	Together with CH433, amplification of the putative minimum promoter region of *kshA2* and its first 21 nucleotides of coding sequence with *Eco*RI-*Nru*I sites, 153 bp
CH392	GGATCCTCTAGA**GAATTC**TGGACGCTCCCGTCTGTTCCGGAA	Tm 55 °C, 30 seg, 30 cycles	Putative promoter of *kshA3* and its first 21 nucleotides of coding sequence with *Eco*RI/*Sca*I restriction sites, 244 bp
CH435	TTTAAACTGCAGAGTACTC**TCGCGA**ACCTGTGCCATCGTCCC		
CH585	GGATCCTCTAGA**GAATTC**CGTGGGACGAAAGTCCTC	Tm 55 °C, 30 seg, 30 cycles	Together with CH433, amplification of the putative minimum promoter region of *kshA3* and its first 21 nucleotides of coding sequence with *Eco*RI-*Nru*I sites, 142 bp
CH593	GGATCCTCTAGA**GAATTC**CTTGATTCGAGACTAGAACACGT	Tm 55 °C, 30 seg, 30 cycles	Together with CH433, amplification of the putative minimum promoter region of *kshA3* and its first 21 nucleotides of coding sequence with *Eco*RI-*Nru*I sites, 213 bp
CH564	GGATCCTCTAGA**GAATTC**CGCGTCTCTCCTGATGTGTCGG	Tm 56 °C, 30 seg, 30 cycles	Putative promoter of *kshB* and its first 21 nucleotides of coding sequence with *Eco*RI/*Nru*I restriction sites, 366 bp
CH565	CTGCAG**TCGCGA**CACCTCGACTGTCGTCATGGCCAGGT		
CH586	GGATCCTCTAGA**GAATTC**TCCCGGTAGCGTCGTGTTGCTGAT	Tm 60 °C, 30 seg, 30 cycles	Together with CH565, amplification of the putative minimum promoter region of *kshB* and its first 21 nucleotides of coding sequence with *Eco*RI-*Nru*I sites, 134 bp

Restriction sites are shown in bold.

**Table 3 microorganisms-09-01171-t003:** Growth of *R. ruber* WT and *ksh* single mutant strains in minimal medium with different steroids as carbon source.

	AcNa 24 mM		CHO 1.6 mM		AD 2.2 mM	
Strain	A_max_	DT (Hours)	A_max_	DT (Hours)	A_max_	DT (Hours)
WT	1.81 ± 0.07	3.57 ± 0.17	1.94 ± 0.11	4.86 ± 0.18	2.20 ± 0.08	2.10 ± 0.10
Δ*kshA1*	2.11 ± 0.05	3.02 ± 0.05	1.65 ± 0.01	4.22 ± 0.09	2.11 ± 0.03	2.64 ± 0.15
Δ*kshA2*	1.72 ± 0.04	3.57 ± 0.03	1.96 ± 0.13	4.74 ± 0.20	2.03 ± 0.02	21.44 ± 1.73
Δ*kshA3*	1.45 ± 0.04	3.73 ± 0.04	1.55 ± 0.03	4.10 ± 0.12	2.04 ± 0.06	2.27 ± 0.09

A_max_, maximal absorbance at 600 nm; DT, doubling time.

## Data Availability

The data presented in this study are available in the Appendix A. qRT-PCR data are available upon request.

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
