# Peer review of "Further Studies on the 3-Ketosteroid 9α-Hydroxylase of *Rhodococcus ruber* Chol-4, a Rieske Oxygenase of the Steroid Degradation Pathway"

_microorganisms, 2021, doi:10.3390/microorganisms9061171_

Round 1

Reviewer 1 Report

In the paper “Further studies on the 3-ketosteroid-9-α-hydroxylase of Rhodococcus ruber Chol-4, a Rieske oxygenase of the steroid degradation pathway”. The authors studied the growth of R. ruber ΔkshA strains on different steroids substrates and the promoter regions of these genes were analyzed. Generally, this manuscript is interesting, meaningful and well planned. It can be accepted after minor revision.

The most important general comments:

  1. Reading the work, one has the impression that it is not fully discussed. I recommend extending the discussion with additional literature items.
  2. In my opinion, the conclusions are extensively described. I believe that it is not necessary to refer to the literature in the conclusions. Please refer to the results of your research in conclusions.
  3. Editorial errors appear throughout the manuscript and supplementary file, eg.: “WT” in Table 3 - unnecessary use of italics; Line 421, 487 - no italics. Please correct the text carefully.

The most important detailed comments:

  1. Figure 3 and 4. Please use the same scale in all graphs.
  2. Please standardize “3-ketosteroid-9-α-hydroxylase” throughout the manuscript.
  3. The quality of the figures is not the best.
  4. Please read the authors' guide carefully and correct references.

Author Response

Dear Reviewer,

We kindly appreciate your time to review our paper and your providing valuable comments that allowed us improving our last version. We provide now a point-by-point response to your concerns. We hope that you find our responses satisfactory and that the manuscript is now acceptable for publication.

Reviewer 1.

The most important general comments:

1. Reading the work, one has the impression that it is not fully discussed. I recommend extending the discussion with additional literature items.

 Answer: We have improved the discussion and included more actual literature. Please see comments to reviewer 2 and also the revised text for more details.

2. In my opinion, the conclusions are extensively described. I believe that it is not necessary to refer to the literature in the conclusions. Please refer to the results of your research in conclusions.

 Answer: We thank the reviewer for his suggestion, we have moved a whole paragraph containing all the references to the proper section for discussion; conclusions are therefore shortened. Please see the revised text for details.

3. Editorial errors appear throughout the manuscript and supplementary file, eg.: “WT” in Table 3 - unnecessary use of italics; Line 421, 487 - no italics. Please correct the text carefully.

 Answer: We are afraid that italics were lost in the submission conversion. We have revised the italics throughout the document, thank you very much for pointing it out.

The most important detailed comments:

  1. Figure 3 and 4. Please use the same scale in all graphs.

Answer: We have resized both Figures using the same scale (Figures 4 and 5 in the actual text)

2. Please standardize “3-ketosteroid-9-α-hydroxylase” throughout the manuscript.

Answer: We have standardized the name as 3-ketosteroid 9α-hydroxylase

3. The quality of the figures is not the best.

 Answer: We have improved the quality and resolution of all the figures in the new document, we expect now that they are suitable for publication.

4. Please read the authors' guide carefully and correct references.

 Answer: References have been introduced using Endnote and attending the journal indications but we are afraid that somehow in the pdf conversion, format was lost. We have copied and revised as simple text the references to avoid this problem.

Reviewer 2 Report

Major concerns:

The authors used Rhodococcus ruber Chol-4 as the model organism to study the essential roles and regulation of ksh genes in steroid metabolism. The genome of strain Chol-4 has been completely sequenced and the candidate gene clusters involved in steroid degradation have been identified in previous studies. In the submitted manuscript, Baldanta et al. tested the steroid substrate spectra of the wild-type and ksh mutants of the strain Chol-4. Moreover, the promoter regions of these genes have been identified through qRT-PCR. Their results show that KshA2B and KshA1B may be involved in AD metabolism, while KshA3B and KshA1B contribute to the cholesterol metabolism in R. ruber. This work provides comprehensive experimental evidence to elucidate the roles of different ksh genes in the biodegradation of different steroid substrates. This study expands our understanding of microbial degradation of recalcitrant compounds, such as cholesterol and androgens. In general, this manuscript is well written and the experiments are well designed. I have only some minor comments:

Minor comments:

  1. By using Rhodococcus ruber as the model organism, the authors described the gene regulation and functional redundancy of ksh genes against different steroid substrates. Actinobacterial Rhodococcus can grow with different steroid substrates, including not only cholesterol and androgens but also bile acids and estrogens. The actinobacterial steroid catabolic pathways are diverse, depending on the structural of different steroid substrates. It is thus important to describe the metabolic diversity and corresponding pathways of Rhodococcus spp. involved in steroid metabolism. Some relevant literature should be included.
  2. The cholesterol degradation pathway is unfamiliar to most readers, thus, it would be nice to show the pathway and the reactions mediated by Ksh enzymes in the Introduction (shown as Figure 1).
  3. Recently, Rhodococcus oxygenase genes and biochemical mechanisms involved in the activation and degradation of estrogenic A- and B-rings degradation have been identified. The phylogenetic and mechanistic difference between the oxygenase genes of androgen and estrogen degradation should be discussed.
  4. The authors concluded that KshA2B and KshA1B are involved in AD metabolism, while KshA3B and KshA1B contribute to the cholesterol metabolism in ruber. However, it appears that the “real” substrates for these different KshAB enzymes are actually androgenic AD or ADD but not cholesterol or other steroids with a side-chain. How can these kshAB genes be induced by different steroids (androgen and cholesterol) but catalyze the ring-cleavage of the same substrates (AD and ADD)? It is interesting and should be discuss in some details.
  5. It would be very nice if the authors test whether the Ksh enzymes can work on estrogenic substrates or bile acids.
